Distribution, characteristics, and importance of particulate and mineral-associated organic carbon in China forest: a meta-analysis

Cheng Hao
Su Yangui
Huang Zhengyi
Lin Sinuo
Yan Jingyi
Wu Guopeng
Huang Gang 307083590@qq.com
School of Geographical Sciences, School of Carbon Neutrality Future Technology, Fujian Normal University , Fuzhou , China
Brygadyrenko Viktor
Electronic publication date: 2025 Mar 26
Publication date: 2025
Volume: 13
Electronic Location ID: e19189
Received 2024 Nov 22; Accepted 2025 Feb 26
Copyright: © 2025 Cheng et al.
Copyright year: 2025
Copyright holder: Cheng et al.
License: This is an open access article distributed under the terms of the Creative Commons Attribution License, which permits unrestricted use, distribution, reproduction and adaptation in any medium and for any purpose provided that it is properly attributed. For attribution, the original author(s), title, publication source (PeerJ) and either DOI or URL of the article must be cited.
License URL: https://creativecommons.org/licenses/by/4.0/

Keywords: Carbon sequestration, Soil carbon fractions, Forest age, Environmental drivers, Soil depth

Funding: National Natural Science Foundation of China 32171643, 41671115, U1703332 This work was supported by the National Natural Science Foundation of China (Nos. 32171643, 41671115, U1703332). The funders had no role in study design, data collection and analysis, decision to publish, or preparation of the manuscript.

==============================
Background

Forest soil organic carbon (SOC) plays a critical role in the global carbon cycle, and increasing long-term forest carbon storage is essential for carbon sequestration. However, the distribution and drivers of mineral-associated (MAOC) and particulate organic carbon (POC) in forest soils at a continental scale remain poorly understood.

Methods

Using 540 data points from 59 studies related to POC, MAOC, and total SOC in China’s forests, we analyzed the distribution of POC and MAOC across forest type, soil depth and soil type, and further investigated their influencing factors.

Results

MAOC accounted for more than 63% of total SOC in forest soils. Both POC and MAOC increase with forest age, with mixed forests showing faster growth compared to monoculture forests. The MAOC/SOC ratio decreases with forest age but increases with soil depth, demonstrating the dominance of MAOC in deeper soils. Importantly, MAOC content continuously increases with SOC, and exhibits no upper limit, suggesting the potential for persistent soil carbon accumulation. MAOC is closely associated to microbial biomass carbon, and POC is mainly related with plant litter biomass.

Conclusion

MAOC and POC are influenced by different environmental factors and display distinct distribution patterns across forest types and soil depths. Thus, differentiating their respective responses to climate change is essential. The carbon sequestration potential of forests in China remains far from saturation.

Introduction

Soil is the largest terrestrial carbon reservoir (Georgiou et al., 2022) and plays an important role in sequestering atmospheric carbon emissions (van Groenigen et al., 2014). C fixation in forests accounts for more than two-thirds of the total amount in terrestrial ecosystems (Fang et al., 2001). In recent years, soil organic carbon (SOC) are usually classified by size into particulate organic carbon (POC) and mineral-associated organic carbon (MAOC). Among them, POC is mainly imported from plant sources, and turns over fast. MAOC is derived from microbial residual input through “dissolved organic carbon (DOC)-microbial route” or physical transfer (mineral adsorption or polymerization process) of particulate organic matter (POM) (Cotrufo et al., 2015). MAOC is the largest, slowest cycling pool of soil carbon (Lavallee, Soong & Cotrufo, 2020; Sokol & Bradford, 2019), and it is considered to be associated with the most critical mechanisms for the long-term stability of SOC (Hemingway et al., 2019). MAOC accounts for more than 50% of the total SOC content in forest soils (Xu et al., 2021). In addition, some studies suggest that POC is very important for soil texture, microbial nutrients, and energy, and is the main part of the soil C component’s response to climate change (Chaplot & Cooper, 2015; Davidson & Janssens, 2006). The variations of POC and MAOC, and their relative contributions to SOC remain unclear yet.

Given the different quantity and quality of litters, climate and vegetation types may affect them differently. Although vegetation productivity is positively associated with litter input, vegetation productivity and soil carbon content are not always closely related in large-scale (Valentini et al., 2000). The reason of this discrepancy have been identified as the different responses of POC and MAOC to the litter quality changes (Castellano et al., 2015; Cotrufo et al., 2013; Tian et al., 2018). However, very little work has been done in large-scale to control POC and MAOC storage by focusing on the effects of climatic factors on forest types and soil properties.

Empirical studies have found that soil C pools and C components were significantly different in forest types, due to their different litter quality and root exudates (Augusto & Boca, 2022). Low litter quality (high C:N ratio and lignin content) and soil acidification in coniferous forests decreased soil enzymatic activities, and thereby can reduce the proportion of MAOC in SOC pools via facilitating POC preservation and decreasing microbial decomposition (Lyu et al., 2023). In particular, for most coniferous trees colonized by arbuscular mycorrhizal fungi (AMF), root exudates may promote the decomposition of MAOC due to the release of mineral elements, and the formation of active minerals (Cotrufo & Lavallee, 2022). To date, The distributions and changes of POC and MAOC in forest types and forest growth are still contradictory (Chen et al., 2023a; Zhang et al., 2024). In addition, soil types (mineral properties) and soil depth act independently and interactively with forest types to determine the POC and MAOC contents and their distribution. For example, mineral properties (kaolinite 1:1 clay minerals, montmorillonite 2:1 clay minerals, and iron or aluminum (oxygen) hydroxide) in soil types have different soil specific surface area and the ability to adsorb and bond organic particles, which are expected to be linked with the proportion of MAOC contents in soil (Kome et al., 2019). Meanwhile, some empirical studies have found that the proportion of MAOC in SOC increases with increasing soil depth, despite microbial biomass carbon (MBC) and microbial activities decrease (Chen et al., 2023b; Han et al., 2017), indicating a dominant role of MAOC form in deep soil (Han et al., 2016). However, few studies have been conducted to determine the distribution patterns of POC and MAOC with soil depth and their interactions with regional climate and vegetation. Therefore, a synthesis of studies that investigate SOC components and their driving factors might help elucidate the mechanisms underlying these divergent effects, and determine the effects of forest management on SOC stabilization.

After long-term afforestation, the forest area in China has reached 220.45 million hectares, and the forest coverage rate is 22.96% (Friedlingstein et al., 2020). Although afforestation is considered a viable mean to enhance terrestrial carbon sinks and reduce carbon dioxide emissions, the long-term C storage capacity and potential C saturation limit of forest soils must be considered in order to sustainably improve forest C capture in the long term. China’s forests cover most of the forest types in the Northern Hemisphere, and the soil carbon storage capacity varies significantly across these forest types. This study focuses on the changes in SOC physical fractions (particulate vs. mineral-associated SOC), and their relative contributions to SOC contents. We collected data related to soil POC, MAOC, and total SOC in forest ecosystems in China. Our specific objectives were : (1) To determine the distribution of POC and MAOC in forest types, forest age, soil types, and soil depths, and their relative roles in SOC; (2) To isolate the relationship between POC and MAOC and climate, plant and soil factors, and further determine their relative importance.

Materials and Methods

Date collection

We used the Web of Science (https://www.webofscience.com), Google Scholar (https://scholar.google.com), and China National Knowledge Infrastructure (CNKI, http://www.cnki.net) to compile a list of all peer-reviewed articles that investigated soil organic carbon components in the forest ecosystem of China. The search keywords were “(Particulate organic carbon OR POC, Mineral-associated organic carbon OR MAOC, Particulate organic matter OR POM), (Mineral-associated organic matter OR MAOM), (Chinese forest), (Soil aggregates)”, and their combinations. To avoid bias in the selection of publications and to increase the comparability of data, we followed the PRISMA guidelines and selected articles that meet the following criteria: (1) the data was directly obtained from field studies in natural forests in China, excluding reviews, modeling studies, and greenhouse experiments; (2) studies focused exclusively on the separation of SOC into physical components. We uniformly selected data using the “sodium hexametaphosphate method” (Cambardella & Elliott, 1992) to determine POC (>53 μm) and MAOC (<53 μm) content, to avoid differences in those with soil texture classification methods; (3) the information on soil C component and forest type must have clearly been obtained. When multiple publications included the same data from one study, the data were recorded only once; (4) the studies included in the synthesis were predominantly from the eastern part of China, covering most of the forest covered areas in China, the data synthesis is not representative enough for the northwest of China and Qinghai-Tibet Plateau. We have neglected some research on atypical forests that are restored from the degraded arid lands. After multiple screenings of articles, we totally collected 540 observational data from 59 independent studies, which cover the major forest ecosystems of China (Figs. 1 and S1).

Figure 1 PRISMA flow diagram showing the procedure used for the selection of studies and data for meta analysis.

The numbers in front of the brackets indicate the number of studies, and the numbers in brackets indicate the number of data pairs.

Apart from the SOC and SOC components, forest types, forest age, sampling depth, climatic properties (mean annual temperature (MAT) and mean annual precipitation (MAP)), vegetation (microbial biomass carbon, litter biomass, living fin root biomass, above-ground biomass carbon) and edaphic properties (soil pH, total organic carbon, total nitrogen, total phosphorus, soil type, dissolved organic carbon, bulk density and Silt+Clay%) in sites for each study were extracted from the materials and method section, tables or supporting information in each study. Meanwhile, we also obtained those parameters of sites from the author’s other literature, based on the author’s name and the site description. We also obtained the coordinates (latitude and longitude) for each study site, and based on them, we derived climatic values from the World Clim database (https://www.worldclim.org) for sites with incomplete information. And we added some parameters from the ISRIC-WISE database (https://data.isric.org) for sites with incomplete information. The interpolated parameters were mainly used to conduct structural equation modeling (SEM) analysis to determine the driving factors and influencing pathways for POC and MAOC.

Based on the soil classification system of the unified FAO UNESCO system (Food and Agriculture Organization of the United Nations (FAO), 2014), we identified eight soil orders (Acrisols, Arenosols, Cambisols, Ferralsols, Lithosols, Luvisols, Chernozems, Vertisols) in this study. Due to the limited data for Arenpsols and Vertisols, we did not conduct compared analysis on soil C components of them.

Due to the majority of sampling depth reporting in the 0–20 cm, we uniformed data at 0-20 cm as generic depth for describing the spatial variations of soil SOC and C components, for minimizing biases of sampling schemes in studies. To describe C components distribution in depth, we scaled the measured values from depth at <20 cm as 0–20 cm. For depth down to 20 to 40 cm depth, we uniformed them as 20–40 cm. We also uniformed data from 40 to 60 cm as 40–60 cm, and data from 60 to 100 cm as >60 cm.

Plant above-ground biomass carbon (ABC, Mg C ha−1) for each plot in this study was extracted from the global above-ground and subsurface standing biomass carbon density dataset developed by Spawn & Gibbs (2020). The dataset is publicly available and can be accessed through the Oak Ridge National Laboratory DAAC Data Repository (https://doi.org/10.3334/ORNLDAAC/1763). These data were then imported into ArcGIS 10.8 to extract above-ground plant biomass carbon from each sample point, using the latitude and longitude values of the sampling points.

In the process of data extraction, it is sometimes impossible to extract MAOC content and POC content directly from the data in the chart. Based on the definitions of POC, MAOC, fine POC (fPOC), coarse POC (cPOC), and previous research experience (Guo et al., 2022; Ghani et al., 2023), we use the following formula for calculation:

(1) POCcontent(g/kg)=cPOCconcentration×cPOCproportion+fPOCconcentration×fPOCproportion,

(2) POCcontent(g/kg)=cPOCcontent+fPOCcontent,

(3) SOCcontent(g/kg)=POCcontent+MAOCcontent.

The literature search and selection process was independently conducted by two authors (Cheng and Huang) to ensure comprehensiveness and consistency. Both authors screened and evaluated eligible studies individually. Throughout the process, no disagreements arose. If any disagreements had occurred, Cheng would have served as the referee to make the final decision; however, this step was not required in the current study.

Statistical analyses

The effects of climatic zones, forest types, soil types and soil depths on the organic carbon compositions (POC, MAOC, SOC, MAOC/SOC) were analyzed using one-way analysis of variance (ANOVA). The independent sample t-test or least significant difference (LSD) test was used to determine the paired differences when possible. Linear mixed-effects models (LMMs) were used to assess the relationships between soil organic carbon components (POC, MAOC), environmental factors (MAT, MAP), soil conditions (bulk density, pH, TN, Silt+Clay%), and biological factors (MBC, ABC, DOC, litter biomass, living fine root biomass), and “study of each observation” was treated as a random effect variable, utilizing the “lme4” package (Bates et al., 2015). Post-hoc multiple comparisons were conducted using Tukey’s honestly significant difference (HSD) test to assess significant differences among different forest types, soil types, and soil depths. Analysis of covariance (ANCOVA) was used to assess whether there were significant differences in the relationships between POC, MAOC, and SOC across different climate zones. The significance of the relationships between soil organic carbon components, environmental factors, soil conditions, and biological factors was assessed at the p < 0.05, p < 0.01, and p < 0.001 levels.

Structural equation modeling (SEM) (piecewiseSEM package) was used to evaluate the direct and indirect associations between environment factors, soil conditions and biological factors, POC and MAOC. A conceptual model of hypothetical relationships was constructed based on prior knowledge, assuming that the response of environment factors indirectly influences soil carbon stocks through soil conditions and biological factors.

Results

Contributions of POC and MAOC to soil C pool in Chinese forests

The POC content in forest soils ranged from 0.18 to 65.33 g/kg, while the MAOC content ranged from 0.61 to 119.68 g/kg, and the MAOC/SOC value ranged from 0.14 to 0.99 (Figs. 2A, 2B and 2D). Forest type had a significant impact on POC and SOC (Table 1), showing higher POC and SOC in mixed forests than in broad-leaf and conifer forests (Figs. 2A, 2C). MAOC accounted for 63% of total SOC across three forest types, showing a constant MAOC/SOC ratio in forest types (Fig. 2D). Both POC and MAOC increased with SOC (Figs. 3A, 3B), but the slope was steeper for MAOC (Fig. 3B, slope: 0.49; p < 0.001) than for POC (Fig. 3A, slope: 0.51; p < 0.001), suggesting the relative dominance of MAOC in SOC composition at high SOC content. Overall, POC, MAOC, and the MAOC/SOC ratio were not significantly affected by either latitude, or above-ground biomass carbon (ABC) (Table 1).

Figure 2 Comparison of soil organic carbon fractions in different climate zones and forest types.

(A) POC content; (B) MAOC content; (C) SOC content; (D) MAOC/SOC. BF, Broad-leaf forest; CF, coniferous forest; MF, mixed forest. The solid circle represents the average value, and the two ends of the solid line represent the 95% confidence interval. Different capital letters indicate significant differences between different forest types (p < 0.05).

Table 1 The effect (p value) of forest age, forest type, latitude, MAT (mean annual temperature) and ABC (above-ground biomass carbon) on soil C fractions, with *, **, and *** denoting a significant effect at p < 0.05, p < 0.01 and p < 0.001, respectively.

		POC (g/kg)	MAOC (g/kg)	SOC (g/kg)	MAOC/SOC	
Forest age	F value	5.38	4.23	6.37	6.33	
	p	<0.01**	<0.01**	<0.001***	<0.001***	
Forest type	F value	3.73	2.16	3.47	2.95	
	p	<0.05*	0.116	<0.05*	0.053	
Age group * Forest type	F value	5.69	4.83	7.82	1.71	
	p	<0.001***	<0.001***	<0.001***	0.126	
Latitude	F value	1.39	1.32	1.64	0.08	
	p	0.243	0.255	0.204	0.777	
MAT	F value	0.34	1.35	0.95	0.00	
	p	0.561	0.249	0.334	0.993	
ABC	F value	2.92	3.10	4.62	0.49	
	p	0.089	0.079	<0.05*	0.486	

Figure 3 Linear relationships between POC and MAOC with SOC (A, B).

The solid line represents a significant linear relationship (p < 0.05), the shaded area represents 95% confidence intervals.

Variation of POC and MAOC during forest growth process

Forest age significantly influenced POC, MAOC, SOC, and the MAOC/SOC ratio, and the interaction of forest type and forest age was also significant (Table 1). When considering each individual forest type, POC, MAOC, and SOC increased dramatically with increasing stand age in mixed forests (Figs. 4A–4C), while those in coniferous forests (CF) and broad-leaf forests (BF) increased relatively slowly. On the other hand, the MAOC/SOC ratio in mixed forests decreased as stand age increased (Fig. 4D), due to the relatively faster rate of POC with stand age compared to MAOC.

Figure 4 Linear relationships between POC, MAOC, SOC, and MAOC/SOC with forest age in different forest types (A–D).

The blue, yellow, and green circles represent mixed forests (MF), coniferous forests (CF), and broadleaf forests (BF), respectively. The solid lines indicate significant linear regressions (p < 0.05), and the shaded areas represent 95% confidence intervals.

Variation of POC and MAOC in soil types

Significant differences in POC, MAOC, and SOC were observed among different soil types (Table 1), while the differences in the MAOC/SOC ratio were not significant (Table 1, p = 0.38, Fig. 5D). Among all soil types, Luvisols exhibited significantly higher MAOC and SOC contents, averaging 21.24 and 31.86 g/kg, respectively. In contrast, Cambisols had significantly lower POC, MAOC, and SOC contents, averaging 3.66, 6.88, and 10.78 g/kg, respectively (Figs. 5A–5C).

Figure 5 Characteristics of soil organic carbon composition in different soil types.

(A) POC content; (B) MAOC content; (C) SOC content; (D) MAOC/SOC. Acr, Acrisols; Cam, Cambisols; Fer, Ferrisols; Lit, Lithosols; Luv, Luvisols; Che, Chernozem. The horizontal line within the box represents the median of the data, and the hollow square represents the mean. The lower whisker extends to the smallest data point greater than or equal to Q1-1.5IQR, and the upper whisker extends to the largest data point less than or equal to Q3+1.5IQR. Different capital letters indicate significant differences between different forest types (p < 0.05).

Distribution of POC and MAOC in soil depth

POC, MAOC and SOC content across forest types decreased with depth, while MAOC/SOC ratio increased (Fig. 6). The MAOC/SOC ratio ranged from 0.70 to 0.83 in deeper soil (>20 cm), suggesting a predominance of MAOC over POC in sub-mineral soil (Fig. 6D). Within the top 20 cm of the soil, the proportion of MAOC to SOC is significantly lower than that in deeper soil. Conversely, within the top 20 cm of soil, POC concentration significantly exceeded that of MAOC (Fig. 6D).

Figure 6 Characteristics of soil organic carbon composition in different soil depths.

(A) POC content; (B) MAOC content; (C) SOC content; (D) MAOC/SOC. The solid circle represents the average value, and the two ends of the solid line represent the 95% confidence interval. Different capital letters indicate significant differences between different forest types (p < 0.05).

Environmental drivers of POC and MAOC

Multiple regressions showed that MAP, LB (litter biomass), BD (bulk density), pH were important in regulating POC content and MAP, LB, BD, MBC, pH and TN (total nitrogen) dominated MAOC content (Table 1). Both POC and MAOC exhibited negative correlations with BD (Figs. 7B, 7E), while positive correlations with MBC and TN. Specially, litter biomass was positively associated with POC (Fig. 7A), but had no effects on MAOC (Fig. 7D). pH showed a highly significant negative linear relationship with MAOC (Fig. 7F), but its relationship with POC was not significant (Fig. 7C). DOC content displayed a positive correlation with MAOC (Fig. 7K), while its relationship with POC is not significant (Fig. 7H).

Figure 7 The relationship between POC and MAOC with soil properties.

(A, D) Linear regression of POC and MAOC with litter biomass, respectively; (B, E) linear regression of POC and MAOC with bulk density, respectively; (C, F) linear regression of POC and MAOC with pH, respectively; (G, J) linear regression of POC and MAOC with MBC, (H, K) linear regression of POC and MAOC with DOC, (I, L) linear regression of POC and MAOC with TN, respectively. DOC, dissolved organic carbon; TN, total nitrogen. The solid lines indicate significant linear regressions (p < 0.05), and the shaded areas represent 95% confidence intervals.

The SEM models explained 60% and 67% variation in POC and MAOC, respectively (Fig. 8). Litter biomass directly influenced POC and was the most important variable in controlling POC contents (Fig. 8A), while MBC and TN directly influenced MAOC, with MBC was the critical factor in controlling MAOC contents (Fig. 8B). Climatic factor (MAP) mainly affected POC indirectly by regulating litter biomass and soil bulk density and pH, while affected MAOC via direct and indirect impacts on MBC and TN (Fig. 8B).

Figure 8 Structural equation modeling (SEM) analysis shows direct and indirect effects of climatic factors (MAP), vegetation (LB) and edaphic factors (MBC, BD, pH, TN) on POC (A) and MAOC (B) pools.

Solid and dashed arrows represent significant (p < 0.05) and non-significant (p > 0.05) paths in a fitted structural equation model depicting impact of variables on the soil C fractions. MAP, mean average precipitation; BD, bulk density; LB, litter biomass, pH; MBC, microbial biomass carbon; TN, total nitrogen.

Discussion

Variations of MAOC and POC in forest types, soil types and depth

Forests play a critical role as potential carbon sinks (Vesterdal et al., 2012), and the carbon sequestration potential varies among different forest types (Cremer, Kern & Prietzel, 2016). Compared to global soils (the global soil average is 0.86–0.89, Gregorich et al., 2006), the relatively lower MAOC/SOC ratio observed in Chinese forest soils suggests a potential limitation in carbon stabilization under local conditions. However, the ratio aligns more closely with global forest systems, which highlights the influence of forest type and management practices on soil carbon dynamics. These findings underscore the importance of regional factors in determining soil organic carbon stabilization.

Consistent with our first hypothesis, both POC and MAOC contents increased with forest age across all three forest types. As forests grew and matured, the contributions from both direct forest input to the soil and the more stable, long-term stored MAOC continued to increase, leading to a sustained rise in SOC with forest age. Long-term forest succession studies across various regions consistently showed that SOC in forest soils could continue to increase with age, with some forests even surpassing 300 years (Xiang et al., 2022). Notably, MAOC also increased during this SOC growth (Zhai et al., 2024). This finding suggests that in the context of climate change, afforestation could indeed serve as an effective carbon sequestration strategy to reduce atmospheric CO2 levels.

In addition, both POC and MAOC were higher in mixed forests than in either broadleaf or coniferous forests. This is consistent with findings from European studies (Cotrufo et al., 2019) and litter addition experiments (Feng et al., 2022), which indicate that litter diversity promotes MAOC accumulation (Xu et al., 2024). One reason could be the more complex species composition and litter inputs in mixed forests. The diversity of species and root structures promotes litter decomposition, increasing POC accumulation (Paula et al., 2021). Meanwhile, the relatively high organic matter diversity and quality (low C:N ratio) in litters of mixed forest would be beneficial for the microbial transformation of plant-derived carbon (Lyu et al., 2023). On the other hand, increased microbial activity in mixed forests by mycorrhizal symbiosis can alleviate microbial nutrient limitations (Ma et al., 2022) and helps boost MAOC accumulation. In addition, with increasing forest age, the MAOC/SOC ratio in mixed forests declines significantly (Fig. 4D), which may suggest a reduction of slow decomposition carbon fraction in old-growth forest. More carbon is stored in the less stable form of POC, making it more susceptible to microbial decomposition. Such shifts could be associated to continuous carbon sequestration capability in old-growth forests.

The MAOC content in Luvisols and Chernozems was significantly higher than in other soil types (Fig. 5B). Luvisols and Chernozems are mainly distributed in temperate and high-latitude regions, where organic matter is relative high, but microbial biomass and microbial carbon use efficiency (CUE) are low (Takriti et al., 2018), because the lower temperatures slow down decomposition and metabolism (Conant et al., 2011). There was no significant difference in the MAOC/SOC ratio among different soil types (Fig. 5D). The reason may be that different soil types contain minerals with similar functions, leading to a comparable degree of organic particle adsorption. Ferralsols, commonly found in tropical/subtropical forests, have strong ionic minerals adsorption capacity, helping to develop iron binding-MAOC (Kleber et al., 2005; Liu et al., 2023). Correspondingly, sierozems are principally developed in northern forest soils, and calcium ions in sierozems facilitate the formation of calcium binding-MAOC (Tang et al., 2023). On the other hand, this variation is related to the intensity of physical adsorption and biological transformation of MAOC in different regions. Tropical/subtropical forests have lower microbial CUE, suggesting that the pathway of ‘microbial-derived’ MAOC might be small, and the formation of MAOC may be primarily determined by clay mineral sorption (direct plant input or after partial decomposed plant residues). In contrast, the higher carbon and nutrient content in temperate forest soils tend to promote microbial growth (with higher CUE) and metabolism, which may be responsible for more MAOC formation (Sokol, Sanderman & Bradford, 2019).

Although clay minerals are more abundant in deeper soils than in surface soils, MAOC decreases with increasing soil depth (Fig. 6B), mainly due to the reduction in dissolved organic matter and microbial biomass (Sokol & Bradford, 2019; Soong et al., 2020). This is supported by the positive correlation between SOC, TN, and MAOC. However, the MAOC/SOC ratio increases with soil depth (Fig. 6D), indicating that the decline in POC is greater than the decline in MAOC as soil depth increases, this may suggest that the distribution and controlling factors of POC and MAOC differ between the organic and mineral soil layers.

Link MAOC with ecosystem C pool components

MAOC constitutes a substantial portion of the SOC pool. As carbon content increases, MAOC will show signs of approaching carbon saturation (Cotrufo et al., 2019). However, in our study, when plotting MAOC against SOC, no distinct inflection point was observed (Fig. 3B). While the upper limit of MAOC in European soils is estimated to be 47 g kg−1 (Lugato et al., 2021), our observations in Chinese forest soils show MAOC levels exceeding this value without reaching saturation. This suggests that forest soils in China may still have the potential to further accumulate MAOC. The content of MAOC and SOC depends on the amounts of clay and silt (Viscarra Rossel et al., 2023). The upper limit of silt+clay percentage in Chinese forest soils reached as high as 92.4% (Fig. S2Q), which may explain why SOC and MAOC in Chinese forest soils did not exhibit an upper limit. Much of China’s secondary forests consist of young forests with high productivity (Wei, Liu & Liu, 2023). The latent soil carbon in these areas could provide significant environmental, social, and economic benefits for China, and contribute to climate adaptation strategies. Recent studies indicated that soil MAOC in Australian forests had also not reached its maximum potential (Viscarra Rossel et al., 2023).

Meanwhile, the interaction between MAOC and other carbon pools highlights the complex dynamic mechanisms underlying soil carbon stability. The stability of MAOC, its interaction with DOC and MBC, and its independence from POC further emphasize its potential as a long-term carbon sink in forest ecosystems.

Controlling factors of POC and MAOC

POC and MAOC share common controlling factors, yet they also exhibit notable differences in their determinants. Based on the analysis of the SEM results, POC was mainly controlled by litter input and soil physical properties, whereas MAOC formation was regulated by more complex biological and chemical factors. POC was typically generated from physically fragmented plant debris, making it more susceptible to changes in external environmental conditions (Si et al., 2024). In contrast to POC, the formation of MAOC was more complex, with MBC and TN playing significant roles in its development. The positive correlation between MBC and MAOC may have supported the notion that microorganisms contributed significantly to MAOC formation through the microbial ‘C pump’ effect (Huang et al., 2019; Liang, Schimel & Jastrow, 2017; Sokol, Sanderman & Bradford, 2019).

We found that the interactive effects of climatic zones and SOC significantly influence POC and MAOC (Table S1), indicating that the combined effects of climatic conditions and SOC content play a crucial role in the formation, distribution, and storage of POC and MAOC. Unfortunately, both MAT and MAP did not show a significant effects on POC and MAOC (Table 1). Similar results have been reported in previous studies, where regionally scaled research indicated that MAOC responses to warming (Cheng et al., 2011) and increased precipitation (He et al., 2012; Rocci et al., 2021; Song et al., 2012) were not evident. MAP and MAT only impacted POC without affecting SOC and MAOC (Rocci et al., 2021), suggesting potential instability of POC and MAOC under global change.

Limitations in our data

The data used in this study primarily come from forests in eastern China. Although forests are predominantly distributed in this region, there is a significant lack of forest soil data from northwest China and the Qinghai-Tibet Plateau, resulting in insufficient geographical coverage. Additionally, most of the data relate only to surface soils, with limited information on deeper soil layers. This limitation may hinder a comprehensive analysis of the interaction between soil depth and soil type. Furthermore, most of China’s current forests are secondary, primarily due to natural regeneration following protection policies. The sustained development of secondary forests may lead to overestimating the accumulation of MAOC and SOC, as well as their linear relationships.

Conclusions

Our findings show that MAOC (mineral-associated organic carbon) and POC (particulate organic carbon) exhibit distinct distributions across forest types and soil depths, as well as different responses to environmental changes. MAOC is primarily driven by microbial biomass carbon, while POC is mainly influenced by litter biomass. MAOC is the dominant component of the SOC pool, with the MAOC/SOC ratio ranging from 61% to 75%. SOC increases with the accumulation of both MAOC and POC, indicating that the carbon sequestration capacity of Chinese forests is far from reaching saturation. The MAOC content in mixed forests is higher than in monocultural forests, suggesting greater resistance to climate change. Overall, our work demonstrates that separating soil fractions enhances our mechanistic understanding of soil carbon reserves and stability, and highlights the importance of distinguishing between the responses of POC and MAOC to climate change.

Supplemental Information

Supplemental Information 1 PRISMA checklist.

Supplemental Information 2 Distribution of study sites used in this research.

Our research includes 21 research sites (purple cycles) in temperate forests and 38 research sites (red circles) in tropical/subtropical forests. Light green represents areas with forest cover, while light blue represents areas without forest cover.

Supplemental Information 3 The relationship between POC and MAOC with soil properties.

(a, f) Linear regression of POC and MAOC with MAT, respectively; (b, g) Linear regression of POC and MAOC with MAT, respectively; (c, h) Linear regression of POC and MAOC with litter biomass, respectively; (d, i) Linear regression of POC and MAOC with living fine root biomass, respectively; (e, j) Linear regression of POC and MAOC with pH, respectively; (k, p) Linear regression of POC and MAOC with bulk density, respectively. (l, q) Linear regression of POC and MAOC with clay+silt%, respectively; (m, r) Linear regression of POC and MAOC with MBC, respectively; (n, s) Linear regression of POC and MAOC with DOC, respectively; (o, t) Linear regression of POC and MAOC with TN, respectively; MAT, mean annual temperature; MAP, mean annual precipitation; MBC, microbial biomass carbon; DOC: dissolved organic carbon; TN: total nitrogen. The solid lines indicate significant linear regressions (p < 0.05), and the shaded areas represent 95% confidence intervals.

Supplemental Information 4 Linear relationships between POC and MAOC with SOC (a, b) in forest soils of different climate regions.

The purple and red circles represent linear regression for temperate forests and tropical/subtropical forests, respectively. The solid line represents a significant linear relationship (p < 0.05), the shaded area represents a 95% confidence interval.

Supplemental Information 5 Effects of POC, MAOC, SOC, temperature zone and their interaction on soil carbon fractions as indicated by P values from ANCOVA.

Supplemental Information 6 Stepwise multivariate regression analyses of soil properties and soil carbon components.

ABC: above-ground biomass carbon; MAP: mean annual precipitation; MBC: microbial biomass carbon; DOC: dissolved organic carbon; LB: litter biomass; ROC: readily oxidized organic carbon; RB: fine root biomass; MAT: mean annual temperature; BD: bulk density; TN: total nitrogen; TP: total phosphorus.

Additional Information and Declarations

Competing Interests

The authors declare that they have no competing interests.

Author Contributions

Hao Cheng conceived and designed the experiments, performed the experiments, analyzed the data, prepared figures and/or tables, authored or reviewed drafts of the article, search Strategy, and approved the final draft.

Yangui Su conceived and designed the experiments, prepared figures and/or tables, and approved the final draft.

Zhengyi Huang performed the experiments, prepared figures and/or tables, and approved the final draft.

Sinuo Lin performed the experiments, prepared figures and/or tables, and approved the final draft.

Jingyi Yan performed the experiments, prepared figures and/or tables, and approved the final draft.

Guopeng Wu performed the experiments, prepared figures and/or tables, and approved the final draft.

Gang Huang conceived and designed the experiments, analyzed the data, prepared figures and/or tables, authored or reviewed drafts of the article, search Strategy, and approved the final draft.

Data Availability

The following information was supplied regarding data availability:

The data is available at Zenodo: Cheng, H. (2024). Research data of “Distribution, characteristics, and importance of particulate and mineral-associated organic carbon in China forest” [Data set]. Zenodo. https://doi.org/10.5281/zenodo.13992794.

The code is available at Zenodo: Cheng, H. (2025). Research code of “Distribution, characteristics, and importance of particulate and mineral-associated organic carbon in China forest: A meta-analysis”. Zenodo. https://doi.org/10.5281/zenodo.14842866.

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
