# Peer review of "Distribution, characteristics, and importance of particulate and mineral-associated organic carbon in China forest: a meta-analysis"

_PeerJ, doi:10.7717/peerj.19189_

## Round 0.1 · original submission · Major Revisions

Dear authors, I ask you to carefully consider each of the reviewers' comments in the new version of your article. I hope that the reviewers will have sufficient grounds to approve this publication.

Reviewer 1 ·

Basic reporting

The authors present an extensive and interesting paper that addresses a significant environmental issue; It is generally well-structured. The introduction is informative. The literature cited is relevant and primarily current. The statistics applied are relevant to and proper to the results obtained in this paper, further reinforcing the confidence in the research. However, I have some constructive feedback to help you enhance the paper.

The abstract does not adequately represent the content of the article, for example the objectives are not clearly stated. Review the objectives, because they are of limited scope. Please rewrite.

The novelty of the study is not clear. The new methods/findings achieved through this study needs to be clearly presented.

Why are some paragraphs in the text highlighted in blue?

Line 137. Based on the soil classification system of the unified FAO UNESCO, we identified eight soil orders (Acrisols, Arenpsols, Cambisols, Ferrarsols, Lithosols, Luvisols, Chernozem, Vertisols). In the FAO UNESCO system there are no soil orders. And there are no Arenpsols or Ferrarsols, but rather Arenosols and Ferralsols. They must also add what year (version) they have used.

Line 143. To describe C components distribution in depth, we scaled the measured values from depth at <20 cm as 0-20cm. For depth down to 20 to 40 cm depth, we uniformed them as 20-40cm. Why did you choose these depths and not the depths of the edaphogenetic horizons?

I recommend not using acronyms in conclusions section.

I wish those changes will contribute to improve your paper

Experimental design

No comment

Validity of the findings

No comment

Additional comments

All abbreviations and acronyms used in tables and figures should be defined in the table notes or figure captions.

Reviewer 2 ·

Basic reporting

no comment

Experimental design

no comment

Validity of the findings

no comment

Additional comments

Please confirm if the font size in the figure meets the publication requirements
The abstract section is concise, and some of the content in the results section should belong to the conclusion, while the conclusion section is too general, for example, it lacks the specific dominant role of MAOC in China's forest SOC pool and how it can continue to increase the capacity of carbon fixation in the future.
line 16 : Reasons why forests soil organic carbon (SOC) is critical to the global carbon budget?
line156 lack of relevant references
line188: the MAOC/SOC value ranged from 0.14 to 99. Please check if this is correct and if “%” is missing.
Some of the results are inconsistent in the relationship between climate factors, biomass and poc and maoc, can they be used to infer the conclusion?
line241-245, the result data is repeated in the description, please put it in the conclusion section.
line265-269: it is mentioned in the conclusion that climate and other factors have an effect on maoc and soc, are the ratios in different stand ages analyzed here based on the same climatic conditions?
The numbering of the icons in the article does not match the text, please revise.

Translated with DeepL.com (free version)

Annotated reviews are not available for download in order to protect the identity of reviewers who chose to remain anonymous.

---

## Round 0.2 · accepted · Accept

Dear authors, I congratulate you on the acceptance of this manuscript for publication.

Reviewer 1 ·

Basic reporting

I recommend publication, my comments were taken into consideration.

Experimental design

I recommend publication, my comments were taken into consideration.

Validity of the findings

I recommend publication, my comments were taken into consideration.

Additional comments

I recommend publication, my comments were taken into consideration.